# Different Taxonomic and Functional Indices Complement the Understanding of Herb-Layer Community Assembly Patterns in a Southern-Limit Temperate Forest

**Mercedes Valerio** [1,*] **, Antonio Gazol** [2] **, Javier Puy** [3] **and Ricardo Ibáñez** [1]

1   Departamento de Biología Ambiental, Facultad de Ciencias, Universidad de Navarra, Calle Irunlarrea 1, 31008 Pamplona, Spain
2   Instituto Pirenaico de Ecología, Consejo Superior de Investigaciones Científicas (CSIC), Avenida Montañana 1005, 50059 Zaragoza, Spain
3   Estación Biológica de Doñana, Consejo Superior de Investigaciones Científicas (CSIC), Avenida Américo Vespucio 26, 41092 Sevilla, Spain
*   Correspondence: mvalerio.1@alumni.unav.es

**Abstract:** The efficient conservation of vulnerable ecosystems in the face of global change requires a complete understanding of how plant communities respond to various environmental factors. We aim to demonstrate that a combined use of different approaches, traits, and indices representing each of the taxonomic and functional characteristics of plant communities will give complementary information on the factors driving vegetation assembly patterns. We analyzed variation across an environmental gradient in taxonomic and functional composition, richness, and diversity of the herb-layer of a temperate beech-oak forest that was located in northern Spain. We measured species cover and four functional traits: leaf dry matter content (LDMC), specific leaf area (SLA), leaf size, and plant height. We found that light is the most limiting resource influencing herb-layer vegetation. Taxonomic changes in richness are followed by equivalent functional changes in the diversity of leaf size but by opposite responses in the richness of SLA. Each functional index is related to different environmental factors even within a single trait (particularly for LDMC and leaf size). To conclude, each characteristic of a plant community is influenced by different and even contrasting factors or processes. Combining different approaches, traits, and indices simultaneously will help us understand how plant communities work.

**Keywords:** environmental gradient; functional traits; habitat filtering; herb-layer; limiting similarity; mixed beech-oak forest; species richness





## 1. Introduction

Given the current threats that biodiversity is facing worldwide [1], and in order to mitigate the impacts of climate change on plant communities [2], it is important to understand how different biotic processes (i.e., plant–plant interactions) and habitat constraints influence vegetation assembly patterns. This is particularly true in rear-edge environments of south Europe due to their isolation and particular adaptations to drier conditions [3]. Mixed beech-oak forests have their southern distribution limit in the north of the Iberian Peninsula where global change is modifying growing conditions and resource availability [4], potentially affecting their structure and functioning [5] as well as threatening their conservation [6]. In temperate forests, the understory hosts most of the vascular plant diversity [7] and its herb-layer regulates many of the forest processes and functions, such as tree regeneration and water and nutrient cycling [5,8,9]. However, despite its importance, our ability to predict understory and, in particular, herb-layer community dynamics is still very limited [10,11].

The study of functional traits is known to be an important complement of the taxonomic approach when studying community assembly [12–14]. In this regard, recent studies

have found that taxonomic and functional understory assembly patterns can show different responses to similar environmental conditions (particularly for traits such as specific leaf area; [15]). Classic theory poses two non-exclusive hypotheses about the main processes underlying community assembly: habitat filtering and limiting similarity hypothesis. The habitat filtering hypothesis [16] states that environmental constraints determine the species that are able to survive or establish at a site, selecting individuals with similar functional characteristics and thus producing convergence in functional traits [17]. The limiting similarity hypothesis [18] argues that competitive interactions among the species of a community filter the potential colonists of a site, favoring the coexistence of species with different resource-use-related characteristics and thus promoting trait divergence [19,20]. Following these hypotheses, trait convergence would be associated with habitat filtering, while trait divergence would indicate limiting similarity. However, it is increasingly acknowledged that differentiating these two processes is not as simple, because in natural communities both processes can affect community assembly simultaneously and also in combination with other processes such as facilitation, dispersal limitation, or historical contingencies [20,21]. In addition, both processes can also give rise to the opposite assembly patterns depending on the nature of the trait that is being considered [22–24].

In this regard, it has been found that each functional trait could be influenced by different environmental factors and processes [2,20,25,26]. For example, traits such as leaf dry matter content (LDMC) and specific leaf area (SLA), despite being negatively correlated [27] and both giving information on the resource foraging strategy of the plant [28], can complement each other, as low LDMC is associated to soil fertility, while high SLA is related to soil fertility but also to shade [29]. By contrast, leaf size and plant height are positively related to competition for light [28,30].

Moreover, even within a single trait, the use of different functional indices can give different information on the factors and assembly processes that are acting on the community. In particular, three different functional indices are often used to study plant communities: functional composition, richness, and diversity. Functional composition is usually measured with the community weighted mean (CWM; [31]) and can be defined as the mean of the trait values in a community. On the other hand, functional richness and diversity measure trait range and dispersion, respectively [25,32]. Functional richness (FRic; [33]) is measured as the convex hull volume, the volume of functional space that is occupied by species irrespective of abundance. Functional diversity can be calculated with the Rao quadratic Index [34], which is the sum of pairwise functional distances between species that are weighted by relative abundance and considers both the functional space that is occupied and the similarity between the most abundant species [35]. Thus, changes in environmental conditions could lead to changes in functional composition (e.g., if the new conditions promote a new mean trait value) while maintaining functional diversity (e.g., if the trait values of the new species that are selected within the community are equally diverse as before; [36]). However, it is still not clear to what extent functional composition and diversity are related [36]. In this regard, describing how taxonomic and functional composition, richness, and diversity of different functional traits are influenced by specific environmental factors and processes would shed light on the main drivers of herb-layer community assembly and favor the application of correct management and conservation actions.

In Europe, southern-limit temperate forests and their understory are of particular interest for the study of community assembly [5]. These forests have different floristic compositions and environmental constraints than their northern counterparts due to the potential influence of xericity on herb-layer vegetation [37] which means that typical temperate species coexist with more Mediterranean ones. In particular, our study was carried out in Bertiz Natural Park, one of the best conserved temperate forests in northern Spain and with a high ecological value [38], where the seldom seen confluence between beech and oak trees can be found [39]. In southern-limit forests we can also expect greater differences between the north-facing and south-facing slopes that create stronger environmental

gradients in terms of solar radiation and soil moisture [7,37]. Regarding radiation, it is expected that the presence of a beech canopy that intercepts the 90% of the incident radiation [40] would make light limiting and select for shade-tolerant species with high SLA [30], while leading to divergence in other leaf traits to avoid light competition [25,28]. Yet, in southern-slopes, the presence of a mixed beech-oak canopy would promote environmental heterogeneity and increase trait divergence in the understory [25]. Regarding soil moisture, it is expected that the southern distribution of these forests would make belowground resources to become more limiting [37], especially in southern slopes, influencing mainly the functional composition and diversity of traits that are related to drought-stress. In this sense, limiting soil moisture might select for smaller leaf sizes [30] or promote leaf size divergence to decrease belowground competition [18]. However, and although it is established that environmental constraints determine understory species composition [41–43], the specific effect of these intermingled environmental factors on the structure and functioning of herb-layer communities among traits and indices is still unclear [37,44].

In our study, we assessed which specific factors drive the taxonomic and functional composition, richness, and diversity of the herbaceous layer in a southern-limit temperate forest. We hypothesize that (i) from a general perspective, the main limiting resources driving herb-layer community assembly in this temperate forest will be light and water availability. (ii) The relationship between the taxonomic and functional changes in the herb-layer will vary depending on the particular functional trait that is being studied. (iii) Each functional trait will be influenced by different environmental factors and processes. (iv) The effect of each factor and process on the forest herb-layer will change depending on the functional index that is used: composition, richness, or diversity; i.e., whether we look at mean values or their variation.

## 2. Materials and Methods

### 2.1. Study Site

This study was conducted in Bertiz Natural Park, a Natura 2000 Special Area of Conservation located in the western Pyrenees (WGS84: 43°9' N, 1°37' W, Navarra, Spain). The climate is oceanic, which favors mild temperatures (mean annual temperature of 14 °C) and high levels of rainfall (mean annual rainfall of 1500 mm uniformly distributed throughout the year; [45]). Our study area (in the *Suspiro* basin) covers 132 ha with steep slopes mainly facing north or south, where elevation ranges from 200 to 600 m. The basin is crossed by a large number of small headwater streams and soils are mostly formed over silicic sandstones and conglomerates of the Bundsandstein-Triasic (see more details in [46]). On average, soils have a pH of 5.0, a 3.8% of organic matter content, and 33.1% content of clay [47]. The basin is covered by a non-managed acidophile and ombrophile forest that is dominated by beech (*Fagus sylvatica* L.) with some dispersed oaks that became abundant in south oriented slopes, especially *Quercus robur* L. [48].

### 2.2. Vegetation Sampling

The study site was divided into 102 cells with a grid of 120 m side length, and within each cell, we randomly selected a sampling plot of 400 m$^2$. The study focused on the forest herb-layer, which is comprised of herbaceous vascular plants and shrubs that are equal or smaller than 1m in height (following [8]). We excluded seedlings and saplings of taller woody species because they do not pass their whole life cycle in the herbaceous layer. The data were obtained from a species inventory that was carried out by the authors in June and July 2016. All vascular plant species that were present in each plot were identified and the percentage cover was estimated. In total, 52 species were sampled (Table S1 in Supporting Information). Nomenclature follows [49].

### 2.3. Trait Measurement

For each species, four plant functional traits that were associated with plant ecological strategies were studied (Table S1; see correlations in Table S2). Plant height was measured

as the size from the ground level to the highest photosynthetically active tissue. Plant height is mainly associated with competitive vigor, but also with whole plant fecundity and plant tolerance or avoidance of environmental stress [27], regulating plant responses to resource availability and disturbance [50]. Leaf size, the surface area of a whole leaf, is negatively related to nutrient stress, drought stress, and high-radiation stress [27]. In addition, it is associated with competition for light [28]. We also measured the specific leaf area (SLA), the area of a fresh leaf divided by its oven-dry mass, and leaf dry matter content (LDMC), which is the oven-dry mass of a leaf divided by its water-saturated fresh mass [30]. Lower values of SLA and higher values of LDMC are related to lower potential relative growth rate or mass-based maximum photosynthetic rate and longer leaf lifespan [27]. These four traits were measured between April and June 2018, on leaves and individuals fully developed, following the protocols provided by [30]. Plant height was measured in a total of 25 individuals per species. For the other three traits, a total of 20 leaves per species (two leaves in ten individuals) were collected, put in sealed plastic bags with moist paper and stored in a fridge so that they remained water saturated until their processing. The leaf area was measured by using a scanner accompanied by the image analysis software ImageJ (v. 1.51; [51]). The leaves were also weighted before and after being dried at 80 °C for 48 h to calculate SLA and LDMC.

## 2.4. Environmental Variables Measurement

We measured ten variables in each of the 102 sampling plots: altitude, slope, leaf litter cover, radiation (mean and variation), moisture (mean and variation), DBH mean (average diameter at breast height), basal area, and *Quercus* basal area (Table S3).

The altitude (in m a.s.l.) was obtained from a topographic map 1:5000 [52] using the GPS coordinates of the permanent plots. The slope (sexagesimal degrees) was measured in the field with a clinometer (Silva Clino Master, Silva Sweden). The topsoil moisture (% vol) and its coefficient of variation (CV) were measured in July 2007 with two soil moisture sensors SM200 (Delta-T device, Cambridge, UK) at a depth of 51 mm. To control within-plot variability and obtain an averaged measure, 20 measurements were made in each plot. The leaf litter cover on the ground (in percentage) was estimated visually in 2016. Radiation and its CV were obtained in 2012 by using hemispheric photography. Photographs were taken at 1 m height, in four points that were distributed regularly within each plot, and using a digital camera (EOS 50D, Canon, Tokyo, Japan) that was equipped with a fisheye lens (4.5 mm F2.8 EX DC, Sigma, Tokyo, Japan). As the photographs were taken on different days, they were standardized and corrected by comparing data with a reference plot on different days and by establishing threshold differences. The images were analyzed with Hemiview software (v. 2.1 SR4; 2009 Delta-T Devices, Ltd., Cambridge, UK), obtaining Indirect Solar Factor (ISF) values. The ISF is an estimation of the indirect radiation levels that were measured at a site against those that were estimated at an equivalent open-air site [40]. As the forest that was studied was a non-managed forest, we assumed that it would be stable enough for us to be able to take environmental measurements in different years without adding important noise to the analyses.

Data for the DBH mean (measured in cm), basal area (in $m^2 ha^{-1}$) and *Quercus* basal area (in percentage), all related to canopy structure, were obtained between July 2016 and February 2018. In each plot, we measured the DBH of the central tree and of the six closest neighbors with a diameter that was greater than 5 cm, so that we measured the trees that were present in more than 80% of the plot area. We calculated the mean DBH of the seven trees that were measured in each plot. We also measured the distance from the central tree to each of the six neighbors and we used the maximum distance that was measured to calculate the total area in which the seven trees were distributed. Finally, we calculated the plot basal area dividing the area of the seven trees that were measured by the area they occupied, and we expressed it in $m^2 ha^{-1}$. The *Quercus* basal area is the percentage of basal area corresponding to oaks (*Quercus robur*, *Quercus petraea*, and *Quercus pyrenaica*).

### 2.5. Calculation of Functional Indices

There were two different data matrices that were created: the first matrix contained the abundances of each species in each plot expressed in percentages, while the second matrix contained the mean trait value across all the individuals that were measured for each species. The trait values for leaf size and plant height were log-transformed in advance to remove skewness. From these two matrices, community functional composition, richness and diversity were calculated for each trait in each plot using the community weighted mean (CWM; [31]), functional richness index (FRic; [33]), and Rao quadratic index [34,35], respectively. The functional indices were calculated using the *dbFD* function of the "FD" package [53,54] in R (v. 3.5.0; [55]). Both FRic and Rao indices may be affected by species richness, which may lead to the appearance of significant relationships between variables simply owing to random variation in species richness [50]. To distinguish random changes from changes that were produced by real assembly processes, null models were created by randomizing (with 999 permutations) species occurrences (for calculating null FRic) and rearranging species abundances among all the species that were present in the species pool of each plot (for calculating null Rao), but in such a way that each plot or community maintained its original species richness and abundance. Randomizations were done with the *randomizeMatrix* function in the "Picante" package [56]. For calculating null FRic, we used presence/absence matrices. Then, differences between the observed index and null index were calculated by using the standardized effect size or SES (i.e., observed–expected/sd(expected); [57]), thus obtaining SESFRic and SESRao indices. Positive values indicate, respectively, greater functional richness and diversity than expected, and negative values indicate lower functional richness and diversity than expected by chance [20]. To avoid bias in the calculation of functional indices [58], we removed plots with three or less species, or, particularly for SESRao, plots without variation in species abundances; 86 plots remained for CWM and SESFRic indices and 83 plots for SESRao.

### 2.6. Data Analysis

To visualize patterns in the herb-layer species composition of the 86 plots that were studied, a non-metric multidimensional scaling (NMDS; [59]) with two dimensions was carried out with the *metaMDS* function in the "vegan" package [60]. For this analysis, a Bray–Curtis distance metric was calculated from the species per plot abundance matrix. We analyzed the relationship between the NMDS plot scores in the first and second axes and the environmental variables by using Pearson's correlation coefficient.

To detect the type of filters or processes that were responsible for the functional composition, richness, and diversity patterns that were found, we tested for relationships between CWM, SESFRic, and SESRao indices for each single trait and the ten environmental variables that were previously mentioned by fitting multiple linear regression models. In these models, each functional index was used as a response variable and the ten environmental variables (altitude, slope, leaf litter cover, radiation and its CV, moisture and its CV, DBH mean, basal area, and *Quercus* basal area) were used as explanatory or independent variables. All of the explanatory variables were previously standardized to zero mean and unit variance to obtain comparable regression coefficients. In order to avoid inflated Type I error rate when testing for relationships between the CWM and environmental variables [61], we performed a maximum (row and column-based) permutation test of significance [62–64], using the *test_cwm* function of "weimea" package in R [65]. Then, for all models (CWM, SESFRic, and SESRao), we carried out a backward model selection process, for which we started with a model containing the ten explanatory variables that were studied and then we removed the non-significant variables one by one. We selected as final model the one which contained only significant variables. Once the models were obtained, we tested for the spatial autocorrelation of the residuals (excepting the models with CWM as response variable, which had undergone permutation and should not have autocorrelation problems). We found that in most cases there was no significant spatial autocorrelation (Figure S1).

We also analyzed the correlation between the environmental variables that were studied by using Pearson's coefficient, which may help with model interpretation. Finally, we analyzed the correlation between the taxonomic and functional composition, richness, and diversity in order to see how taxonomic changes influenced community functioning. Again, to avoid Type I error that was related to the CWM approach [61], we calculated the correlation between taxonomic and functional composition using a column-based permutation test (modified permutation test in [66]) with the *test_cwm* function of "weimea" package in R [65]. For richness and diversity we used ordinary Pearson's correlation.

## 3. Results

### 3.1. Patterns in Taxonomic Composition and Richness

The first axis of the NMDS that was carried out to visualize the patterns in the herb-layer species composition differentiated communities from sites with higher values of *Quercus* basal area and light availability (i.e., radiation) from communities of sites with higher moisture and leaf litter cover (Figures 1 and 2; Table S4). According to the second axis, sites with higher light availability and heterogeneity (i.e., higher CV for radiation) and higher *Quercus* basal area accommodated communities with higher species richness than sites with higher altitude, moisture, and leaf litter cover (Figures 1 and 2; Table S4). Light availability was the variable that was most highly correlated with both axes (Figures 1 and 2; Table S4).

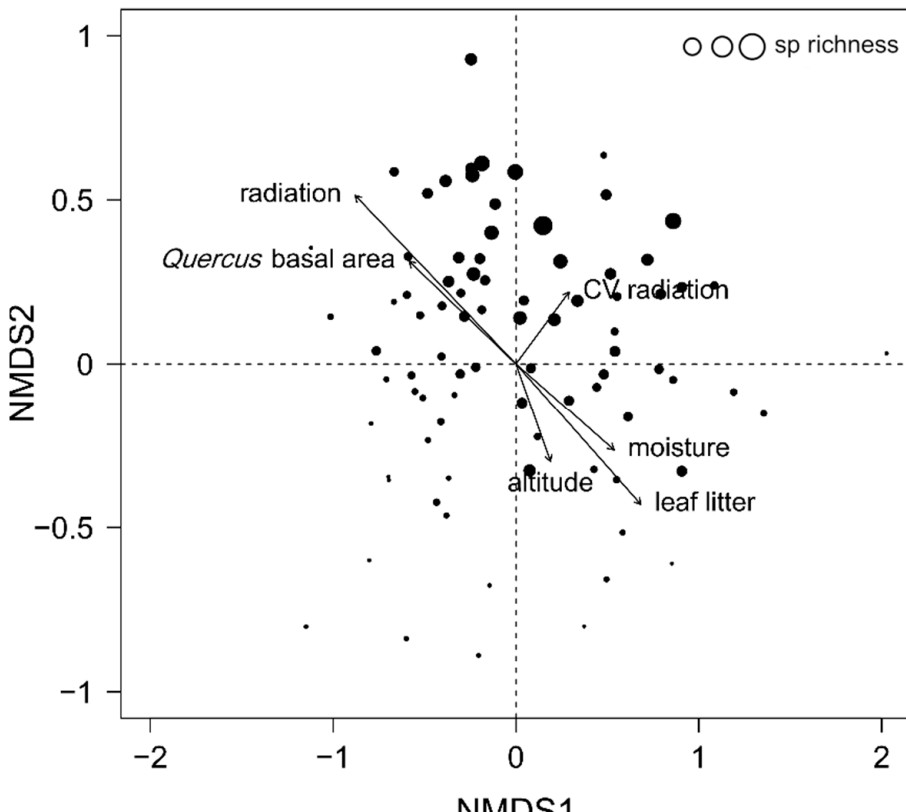

**Figure 1.** NMDS diagram of the herb-layer species composition (stress = 0.2; iterations = 95) and significant relationships between species composition and environmental variables such as altitude, leaf litter cover, moisture, radiation and its coefficient of variation (CV), and *Quercus* basal area (arrows). Other environmental variables were not significant. The length of each arrow indicates the correlation value. Circles represent each of the 86 plots of the basin, with size being proportional to the species richness.

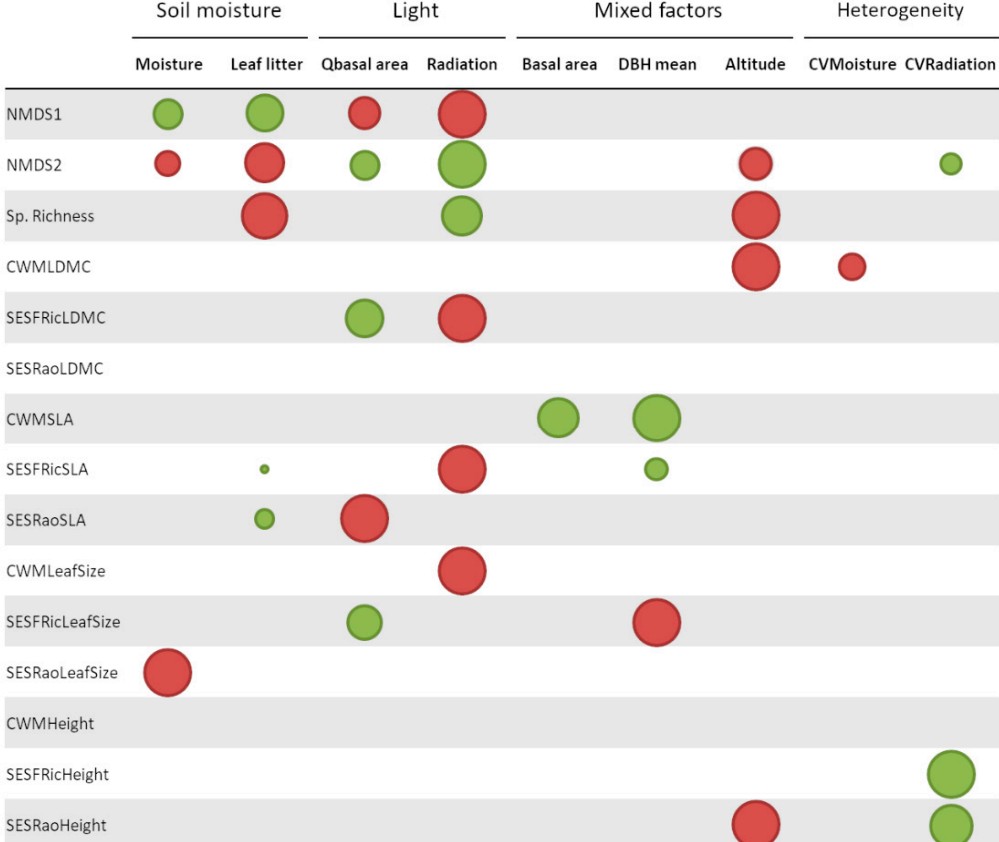

**Figure 2.** Relative influence of environmental factors (in columns) on different characteristics of the herb-layer. Such characteristics are taxonomic (first and second ordination axes scores, herb-layer species richness) and functional indices (CWM: functional composition; SESFRic: standardized effect size of functional richness; SESRao: standardized effect size of functional diversity) for LDMC (leaf dry matter content), SLA (specific leaf area), leaf size, and plant height. The environmental factors are classified according to the main resource to which they are more related (soil moisture, light, mixed factors, heterogeneity). The circle size is proportional to the values of correlation or regression coefficients for each model, while color indicates the sign of the relationship (red: negative relationship, green: positive relationship).

### 3.2. Patterns in Functional Composition, Richness, and Diversity

Regarding LDMC (Table 1; Figures 2 and S2a), the functional composition (CWM) was lower at sites with higher altitude and higher soil moisture variability. By contrast, the functional richness of LDMC was significantly higher at sites with lower radiation but with higher *Quercus* basal area. The functional diversity of LDMC was not significantly influenced by any environmental factor. In terms of SLA (Table 1; Figures 2 and S2b), the CWM was higher at sites with higher basal area and high-diameter trees (i.e., DBH mean). Similarly, the functional richness of SLA was higher at sites with lower radiation, higher leaf litter cover, and high-diameter trees. The functional diversity was higher at sites with higher leaf litter cover and lower *Quercus* basal area. In relation to leaf size (Table 1; Figures 2 and S2c), higher CWM values of leaf size were found at sites with lower radiation. By contrast, the functional richness was higher at sites with higher *Quercus* basal area and with low-diameter trees, and the functional diversity was higher at drier sites. Finally, regarding plant height (Table 1; Figures 2 and S2d), although the CWM was not significantly related to any environmental factor, the functional richness was found to be higher at sites with heterogeneous levels of radiation. Similarly, the functional diversity of plant height was higher at sites with heterogeneous levels of radiation and lower altitude.

**Table 1.** Results of the selected linear regression models testing for relationships between functional composition (CWM), standardized effect size of functional richness (SESFRic), and diversity (SESRao) of the four traits that were studied (leaf dry matter content, specific leaf area, leaf size, and plant height), and environmental variables. The values that are presented in the table are the estimate with the standard error, the F value, the *p*-value that is associated with the F-test and the degrees of freedom of each model (df). Significant relationships ($p < 0.05$) are highlighted in bold. '.' = $p > 0.05$; '*' = $0.01 < p < 0.05$; '**' = $0.001 < p < 0.01$; '***' = $p < 0.001$.

| Trait | *Index* | Predictor | Estimate | Standard Error | F Value | *p*-Value | df |
|---|---|---|---|---|---|---|---|
| **Leaf Dry Matter Content (LDMC)** | | | | | | | |
| | *CWM* | | | | | | 83 |
| | | altitude | −10.781 | 2.556 | 17.786 | **0.012 \*** | |
| | | CV moisture | −6.029 | 2.736 | 4.856 | **0.018 \*** | |
| | *SESFRic* | | | | | | 83 |
| | | radiation | −0.281 | 0.104 | 4.041 | **0.048 \*** | |
| | | *Quercus* basal area | 0.224 | 0.104 | 4.618 | **0.035 \*** | |
| | *SESRao* | | | | | | 81 |
| | | altitude | −0.130 | 0.066 | 3.826 | 0.054. | |
| **Specific Leaf Area (SLA)** | | | | | | | |
| | *CWM* | | | | | | 83 |
| | | basal area | 0.865 | 0.407 | 4.506 | **0.038 \*** | |
| | | DBH mean | 1.020 | 0.403 | 6.404 | **0.032 \*** | |
| | *SESFRic* | | | | | | 82 |
| | | radiation | −0.402 | 0.106 | 15.626 | **0.0002 \*\*\*** | |
| | | leaf litter | 0.063 | 0.106 | 8.866 | **0.004 \*\*** | |
| | | DBH mean | 0.188 | 0.091 | 4.262 | **0.042 \*** | |
| | *SESRao* | | | | | | 80 |
| | | leaf litter | 0.080 | 0.062 | 5.549 | **0.021 \*** | |
| | | *Quercus* basal area | −0.201 | 0.062 | 10.385 | **0.002 \*\*** | |
| **Leaf size** | | | | | | | |
| | *CWM* | | | | | | 84 |
| | | radiation | −0.355 | 0.064 | 31.234 | **0.006 \*\*** | |
| | *SESFRic* | | | | | | 83 |
| | | *Quercus* basal area | 0.306 | 0.123 | 6.180 | **0.015 \*** | |
| | | DBH mean | −0.417 | 0.123 | 13.518 | **0.0004 \*\*\*** | |
| | *SESRao* | | | | | | 81 |
| | | moisture | −0.339 | 0.079 | 18.343 | **0.0001 \*\*\*** | |
| **Plant height** | | | | | | | |
| | *CWM* | | | | | | 84 |
| | | slope | 0.044 | 0.015 | 8.206 | 0.208 | |
| | *SESFRic* | | | | | | 84 |
| | | CV radiation | 0.292 | 0.109 | 7.190 | **0.009 \*\*** | |
| | *SESRao* | | | | | | 80 |
| | | altitude | −0.313 | 0.099 | 8.297 | **0.005 \*\*** | |
| | | CV radiation | 0.281 | 0.099 | 7.987 | **0.006 \*\*** | |

*3.3. Taxonomic vs. Functional Composition, Richness, and Diversity*

We found significant relationships between the functional indices and herb-layer species composition, while no significant relationships were found with species richness (Table S5). The NMDS1 axis was significantly and positively correlated with the CWM of leaf size, functional richness and diversity of SLA, and the functional diversity of plant height (Figure 3a,c,d,f; Table S5). By contrast, it was negatively correlated with the functional richness of leaf size and functional diversity of LDMC (Figure 3b,e; Table S5). The NMDS2 axis was negatively correlated with the functional richness of SLA and positively correlated to the functional diversity of leaf size (Figure 3g,h; Table S5).

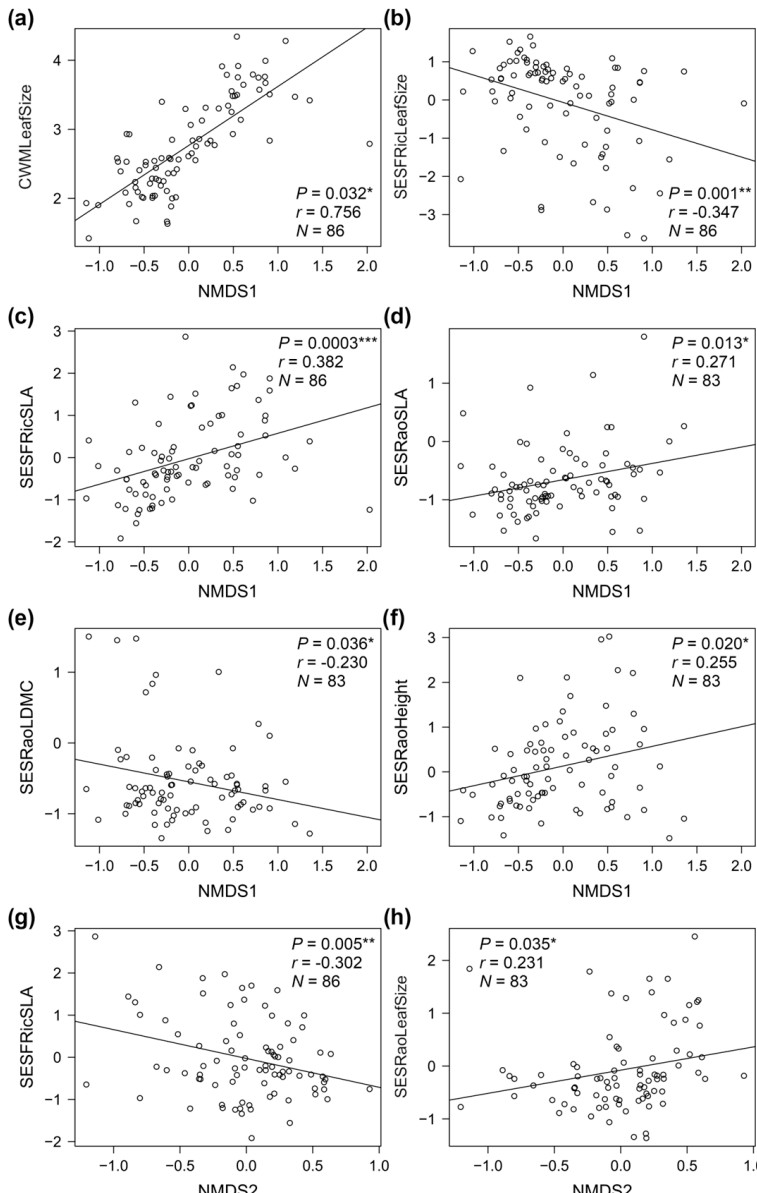

**Figure 3.** Scatter plot representing the significant relationships that were found between the functional composition (CWM), standardized effect size of functional richness (SESFRic) and diversity (SESRao) of the traits that were studied (LDMC: leaf dry matter content, SLA: specific leaf area, leaf size, and plant height), and taxonomic composition (expressed as the first and second ordination axes scores that were given by the NMDS) (**a–h**). The regression line and *r* (Pearson's correlation coefficient), N (number of plots), and *p*-values (statistical significance of the correlation) are shown. '.' = $p > 0.05$; '*' = $0.01 < p < 0.05$; '**' = $0.001 < p < 0.01$; '***' = $p < 0.001$.

## 4. Discussion

In line with our first hypothesis, the main limiting resources that were acting over the forest herb-layer were light and soil moisture. In general, we found a stronger influence of light rather than soil moisture on both the structure and functioning of the herb-layer of this rear-edge temperate forest (Table 1; Figure 2), either because soil moisture is not as limiting as light, or because the herb-layer vegetation that is present in these forests tolerates certain levels of water scarcity [37]. In line with our second hypothesis, changes in the herb-layer taxonomic composition and richness along an environmental gradient were followed by equivalent functional changes in the case of functional diversity of leaf size, but by opposite changes in the case of functional richness of SLA [15] (Figure 3). Regarding

the third and fourth hypotheses, the relative importance of each factor and process that was acting over the herb-layer was dependent on the specific trait that was studied [25] and even on the index that was used within each trait [36], particularly in the case of LDMC and leaf size (e.g., depending on the index that was used, leaf size can be related to light, canopy variables, or soil moisture). In other words, most of the traits and indices that were studied give non-repetitive, complementary information on the main factors influencing herb-layer communities (Figure 2). This supports that in order to acquire a complete understanding of herb-layer community assembly, various indices and traits should be used. Moreover, by describing the specific environmental factors influencing the taxonomic and functional composition, richness, and diversity of each trait, we have disentangled that the maintenance of heterogeneous environments would be key for maintaining the observed taxonomic and functional diversity levels in the herb-layer [43]. However, it is also true that the environmental variables that were used in this study were collected in different years, and year-to-year variation in environmental variables can affect the ability to detect their effect on vegetation patterns. Although in our case we focus on the general patterns and on the relative variation of these variables across space, which might not be significantly influenced by year–year variation, we suggest that a more systematic characterization of the environmental conditions in the study site may reinforce the results that were found in this study.

### 4.1. Patterns in Taxonomic Composition and Richness

The relationship that was found between the first ordination axis, and radiation and *Quercus* basal area is consistent with the well-known effect of light availability and canopy composition on the herb-layer species composition of temperate forests [41,67–70]. The dominance of an oak canopy or at least the presence of a mixed canopy tends to be related to higher light availability in the herb-layer (Table S6), as oaks are more frequently found in southern zones and its canopy intercepts less light than beech [37,48,71]. Thus, the left side of the ordination shows an herb-layer species composition that is associated with higher levels of light in the understory or aboveground resource availability. These conditions favored the presence of species such as *Agrostis capillaris*, *Avenella flexuosa*, *Calluna vulgaris*, *Cytisus scoparius*, *Erica vagans*, *Festuca rubra*, and *Pteridium aquilinum* (in line with [41]). On the contrary, the right side of the ordination seems to be signalling an herb-layer species composition that is characteristic of sites with higher soil or belowground resource availability, such as higher soil moisture or higher leaf litter cover, with species such as *Athyrium filix-femina*, *Dryopteris affinis*, *Helleborus viridis* subsp. *occidentalis*, *Polystichum setiferum*, and *Scilla lilio-hyacinthus*. This is consistent with previous studies showing an influence of soil moisture and leaf litter on species composition [72,73]. In this regard, leaf litter cover is positively related to soil moisture [74] (Table S6) but also to nutrient availability [74]. Thus, these results suggest that the taxonomic composition of the forest herb-layer is influenced both by above and belowground resource availability, and in opposite ways.

When focusing on the second ordination axis, we found almost similar patterns as for the first axis. But in this case, the herb-layer communities that were located at sites with higher light or aboveground resource availability and heterogeneity (i.e., positive values of second ordination axis), tended also to have higher species richness than sites with higher moisture or soil resource availability. The positive effect of light and of a mixed canopy on herb-layer richness has also been reported by other authors [43,44]. The negative influence of altitude on species richness might be related to the habitat filtering process that is caused by different environmental stresses or by processes such as dispersal limitation [75,76]. Sites with higher leaf litter cover and moisture tend to have lower herb-layer species richness, as a thick leaf litter layer, despite increasing water availability (Table S6), can impede seed germination through light interception [74,75]. Thus, at least from a taxonomic approach, light availability seems to have a stronger influence over herb-layer composition than soil resource availability.

*4.2. Patterns in Functional Composition, Richness, and Diversity*

According to the regression models, the functional composition (CWM) or mean value of LDMC was higher at low altitude sites with homogeneous levels of soil moisture (Table 1). Some of the species that were present at these sites were *Blechnum spicant*, *Daphne laureola*, *Hedera helix*, and *Ruscus aculeatus*, which are species that are typical of shaded and humid sites [37,49]. In addition, low altitude sites might also be related to lower levels of disturbance (i.e., tree fall by wind; [37,41,77]). Thus, it seems consistent that lower sites which are less exposed to wind and have enough water availability favor the presence of species with long lifespan [78], which tend to have higher LDMC [30]. By contrast, the functional richness and diversity of LDMC were not influenced by the same factors as functional composition (in line with [36]). The functional richness of LDMC was lower at sites with higher radiation and with a pure beech canopy. These conditions can be found in large canopy gaps that were opened in north-facing slopes, where there is also a pure beech canopy [75]. At these sites, gap formation might cause a huge increase in productivity levels [79], which might lead to a process of competition for light (stronger at sites with high resource availability; [5]), and end in the exclusion of less competitive species [80], thus reducing the functional richness of LDMC [81]. Finally, the functional diversity of LDMC tends to be higher at negative values of the first ordination axis, in communities with species such as *Agrostis capillaris*, *Calluna vulgaris*, or *Pteridium aquilinum*, which might probably have diverse values of LDMC. However, the environmental factors influencing LDMC diversity seem to be different from those that were studied and influencing species composition.

In terms of SLA, we found higher mean values of SLA (i.e., species with large and thin leaves) at denser forest stands with large-diameter trees. The higher SLA might be a result of the higher productivity levels that were found at these sites, as previous studies have related basal area and SLA with soil fertility [25,28,30]. However, SLA is a complex trait that can also be influenced by shade [29]. This increase in the functional composition or mean value of SLA is related to the functional richness of SLA, which was higher at shaded sites with large-diameter trees and higher leaf litter cover, in line with results that were found in previous studies [73]. Large-diameter trees and leaf litter are indicators of non-disturbed, high-quality sites, with higher resource availability and productivity levels [28,74,82]. Thus, this result is consistent with the divergence in leaf traits that has been reported at sites with higher productivity levels but also with a resulting high competition for light [25]. A similar explanation can be given for the higher functional diversity of SLA that was found at sites that were dominated by beech and with higher leaf litter cover. However, the higher richness and diversity of SLA can also be explained by the microclimatic buffering that is happening in dense and shaded sites of old-growth forests [83], in which less competitive forest specialist species are favored [15,70,84]. These results are consistent with the relationships that were found between the functional richness and diversity of SLA and the two ordination axes, showing that the higher richness and diversity of SLA appeared in communities that were located at shaded sites with higher leaf litter cover and that host a low number of species such as *Athyrium filix-femina*, *Helleborus viridis* subsp. *occidentalis*, and *Scilla lilio-hyacinthus*. Moreover, this suggests that the taxonomic richness and functional richness of SLA might have opposite responses to similar environmental factors. In this regard, previous studies have reported lower species richness but higher functional diversity at sites with strong microclimatic buffering [15].

Regarding leaf size, the lower mean values of leaf size (i.e., smaller leaves) that were found at sites with higher radiation is consistent with the generalized idea that small leaves are an adaptation to high-radiation stress [30]. In this sense, some of the species that were found in communities with higher radiation levels were *Calluna vulgaris*, *Daboecia cantabrica*, and *Erica vagans*, which is consistent with previous studies finding *Ericaceae* at xeric sites [37]. These species that were found at sites with higher radiation tended to have lower values of leaf size, as shown by the positive relationship between the functional composition or the mean value of leaf size and the NMDS1 scores. By contrast, the functional richness of leaf

size was not directly related to radiation, but to sites with oaks and small-diameter trees. Oaks tend to grow in southern zones and canopy openings [85,86] and the presence of a mixed canopy of beech and oak allows light in the herb-layer [48] (Table S6). Moreover, small-diameter trees are related to higher light heterogeneity (Table S6). Thus, higher light availability or heterogeneity might favor the coexistence of species with different leaf sizes, increasing the functional richness [25]. Although these sites can have high light availability, other type of resources, such as water, can be scarce. In this sense, small-diameter trees are also related to drought [87] and thus, to a stronger competition for water [73]. This detail might explain the results that were found for the functional diversity of leaf size, which was higher at drier sites. Thus, the higher diversity that was found in these sites could be a result of a limiting similarity process [18,19] to avoid competition for water [85]. Regarding the relationship between the taxonomic and functional approach, we found that the communities that were present at sites with a mixed canopy and low soil moisture, and which tended to have higher species richness, also had higher richness and diversity of leaf size (probably due to the presence of species with large differences in leaf size such as *Calluna vulgaris* or *Pteridium aquilinum*). Thus, this might indicate that taxonomic richness and functional diversity of leaf size show equivalent responses to similar environmental factors.

The mean value for plant height was not influenced by any of the studied environmental factors. By contrast, the functional richness of plant height was higher at sites with higher light heterogeneity, which is consistent with previous studies suggesting environmental heterogeneity as a source of trait divergence [22,25], and with the known relationship between plant height and light [28]. Similarly, the functional diversity of plant height was higher at lower sites with heterogeneous levels of light. The reason for finding higher plant height diversity at lower sites could be the lack of stress by wind or other factors, which allows the presence of species with different heights [20,88]. This might be consistent with the positive relationship that was found for plant height diversity and communities consisting of species such as *Helleborus viridis* subsp. *occidentalis*, *Polystichum setiferum*, and *Scilla lilio-hyacinthus*.

These results show how taxonomic composition and richness patterns are related in different ways to each functional index. The taxonomic composition influenced the diversity of LDMC and the mean value and richness of leaf size, on the one hand (with heliophyte species having diverse values of LDMC, different values of leaf size, and low leaf size mean values), and the diversity of SLA and plant height, on the other hand (with shade-tolerant species having diverse values of SLA and plant height). In addition, the second ordination axis that was more related to species richness, also seems to be related to certain functional indices in opposite ways depending on the trait being studied (leaf size or SLA), with communities with higher species richness having also higher diversity of leaf size, and communities with lower species richness having higher richness of SLA. In addition, each of the functional indices is related to different factors and processes acting over the forest herb-layer, even within a single trait. This is particularly true in the case of LDMC and leaf size, while for plant height and SLA there are more similarities in the information that is given by functional richness and diversity indices. Thus, it is important to use different indices and traits within the functional approach, and to combine it with the taxonomic approach to obtain a clearer understanding of herb-layer community assembly.

## 5. Conclusions

Light availability rather than soil moisture is the most limiting resource influencing the structure and functioning of herb-layer communities in this rear-edge temperate forest. However, the relative importance of each environmental factor and assembly process over the different dimensions of the herb-layer differs between approaches (taxonomic and functional), traits (LDMC, SLA, leaf size, and plant height), and indices (composition, richness, and diversity). Thus, in order to obtain a broader insight of the factors and processes driving plant community assembly, a combination of taxonomic and functional compo-

sition, richness, and diversity indices should be used. In addition, as the environmental variables that were used in this study were measured in different years and this can lead to a potential bias in the relationships that were detected, a systematic characterization of the environmental conditions should be conducted to reinforce the results that were found. This way, we will be able to understand the specific environmental conditions influencing the different taxonomic and functional characteristics of plant communities and, in the face of global change, apply useful conservation measures which account for these various dimensions of plant communities simultaneously.

**Supplementary Materials:** The following supporting information can be downloaded at: https://www.mdpi.com/article/10.3390/f13091434/s1, Table S1: List of the species that were studied, with their abundance, life form, and mean value of the functional traits that were measured; Table S2: Pearson's correlation values between the functional traits that were studied; Table S3: List of the environmental variables that were studied with their range, mean, and unit; Table S4: Pearson's correlation values between the environmental variables and the axes scores of the taxonomic ordination; Table S5: Pearson's correlation values between taxonomic and functional indices; Table S6: Pearson's correlation values between the environmental variables that were studied; Figure S1: Spatial auto-correlation plots; Figure S2: Relationships between the functional indices and the most influential environmental factors in each case.

**Author Contributions:** Conceptualization, A.G. and R.I.; Formal analysis, M.V.; Funding acquisition, M.V., A.G., J.P. and R.I.; Methodology, M.V., A.G., J.P. and R.I.; Supervision, A.G. and R.I.; Writing—original draft, M.V.; Writing—review & editing, A.G. and R.I. All authors have read and agreed to the published version of the manuscript.

**Funding:** This research was funded by FUNDACIÓN CAJA NAVARRA, grant number 10833 (Program "Tú Eliges, Tú Decides") and UNIVERSIDAD DE NAVARRA (projects "Biodiversity Data Analytics and Environmental Quality" and "Red de Observatorios de la Biodiversidad de Navarra (ROBIN)"). M.V. was supported by DEPARTAMENTO DE EDUCACIÓN, GOBIERNO DE NAVARRA (Ayudas predoctorales para la realización de programas de doctorado de interés para Navarra; Plan de Formación y de I + D 2018). A.G. was supported by the Spanish Ministry of Science and Innovation, grant number RyC2020-030647-I, and by CSIC, grant number PIE-20223AT003. J.P. was supported by the Spanish Ministry of Science and Innovation and EU "NextGenerationEU/PRTR", grant number FJC2020-042954-I.

**Data Availability Statement:** The data that are presented in this study will be openly available in Dryad Digital Repository at https://doi.org/10.5061/dryad.4j0zpc8ds (accessed on 1 September 2022).

**Acknowledgments:** We are grateful to the Parque Natural "Señorío de Bertiz" for allowing us to conduct this research within the protected area. We thank Jon Miguel, María Morán, and Mara Equisoain and other undergraduate students for their help in field work, and Xavier Picó and Lars Götzenberger for their comments on the manuscript.

**Conflicts of Interest:** The authors declare no conflict of interest. The funders had no role in the design of the study; in the collection, analyses, or interpretation of data; in the writing of the manuscript; or in the decision to publish the results.

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
