# Peer review of "Different Taxonomic and Functional Indices Complement the Understanding of Herb-Layer Community Assembly Patterns in a Southern-Limit Temperate Forest"

_forests, doi:10.3390/f13091434_

Round 1

Reviewer 1 Report

Dear Authors,

This is an article dealing with the environmental factors driving vegetation patterns of the herb-layer of a beech-oak forest in northern Spain. Different indices, describing the functional composition, richness, and diversity of the forest herb-layer, were calculated from four functional traits of the herbaceous species.

Although this is a well-presented MS, with a notable number of plots, unfortunately there is a major methodological issue. Data used in the analyses are from different time frames; species survey is from 2016, traits measurements from 2018, moisture from 2007, radiation from 2012, litter from 2016 etc.

I understand that most (if not all) of the species are perennials, and one can assume that the subfloor plant community is the same before and after the survey, but this a weak and undocumented assumption. Additionally, the environmental factors considered are not steady every year, they are not steady even between seasons. Another assumption would have been that the environmental factors set is describing the “mean environment” of the area, and one can relate the mean conditions of the site to the plant community. But the set is referring not only to a different year comparing to species survey, but to totally different years between the various environmental factor measurements. Inter-annual variation and dependence between factors (e.g., radiation-soil moisture, radiation-species coverage, radiation-traits) cannot be overcome.

I’m afraid I see a serious methodological flaw here and I have to suggest a rejection to this MS.

Kind regards

Reviewer 2 Report

The authors try to assess which are the main environmental drivers of taxonomic and functional composition and diversity patterns of the herb-layer of a 18 temperate beech-oak forest located in northern Spain. I find the idea of the ms interesting. The ms in a few cases needs revision regarding the English language (I am not a native speaker myself, but I would suggest checking again the language). I find the references used, quite updated and relevant for the study. Nevertheless, I think that there are a few methodological shortcomings, which I describe below. 

Major concerns

I would like to formulate some questions – methodological shortcomings to which I concluded after going through the manuscript. 

a)     It is not clearly explained from the beginning of the manuscript what is functional composition and what is the meaning of this term? Also, the authors have to state clearly throughout the text what type of patterns they describe (assembly patterns, diversity patterns). In my opinion all these terms: composition, assembly, diversity, richness are mixed and the reader is getting confused till the end of the ms.

b)     In the discussion section, in line 347 you state “different indices and traits should be simultaneously studied”. FRic and RaoQ indices offer a great opportunity to apply a multi-trait analysis. Instead of that you use these indices only for single trait analyses. Please explain.

c)     According to the statistical analyses you have followed, you have not included “space”, spatial autocorrelation anywhere? Is it possible to include space applying other models (glmm, gls) instead of linear multiple regression or using Pearson’s correlation coefficient? 

Minor comments

Line 21: taxonomic and functional ? patterns (what type of patterns)?

Lines 32-34: Large sentence. It could be better to split it.

Lines 59, 62: First, because….Second, because …maybe you should rewrite this

Lines 73-74: This is sentence is not clear to me

Line 80: Please explain what do you mean with the term functional composition

Line 112-113: This sentence should be re-written

Line 218: global?

Lines 273-274: How this sentence is related to the graph?

Lines 284-285: I am not sure if this the best way to present statistical results, it is confusing.

Lines 320-322: I think these lines have to be deleted.

Lines 359, 363: a herb-

Reviewer 3 Report

This article makes a significant contribution to the existing knowledge on the issue in question. The authors have handled the topic very well and prepared the conceptual work in a form that arouses the reader's interest.

In the introductory part, the elements that introduce the problem with clearly defined research objectives are listed.

Material and methods are clearly presented. all elements are clearly described, from the research area to sampling methods to data processing.

The results are clear, comprehensible, precise, to the extent they should be, neither too broad nor too narrow.

The conclusion is clear and fully connected with the hypothesis set at the beginning of the research and is based on the results.

Overall, a very good scientific research article.
